# Effects of Air Anions on Growth and Economic Feasibility of Lettuce: A Plant Factory Experiment Approach

Sora Lee [1] 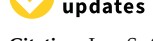, Min-Jeong Song [2] and Myung-Min Oh [2,3],*

1 Department of Agricultural Engineering, National Institute of Agricultural Sciences, Jeonju 54875, Republic of Korea
2 Division of Animal, Horticultural and Food Science, Chungbuk National University, Cheongju 28644, Republic of Korea
3 Brain Korea 21 Center for Bio-Health Industry, Chungbuk National University, Cheongju 28644, Republic of Korea
* Correspondence: moh@cbnu.ac.kr; Tel.: +82-43-2612530

**Abstract:** Anions are molecules that have gained one or more extra electrons, and oxygen anions are the anions most commonly present in the atmosphere. Several studies have reported an improvement in plant respiration and growth through the application of air anions in several plants. In this study, the effect of air anions on the growth of lettuce was explored, and further, the economic feasibility of this technique was analyzed in a plant factory. Two cultivars of lettuce were cultivated for 4 weeks with the application of negatively ionized air in a commercial plant factory. The exposure to air anions improved the growth of the lettuce plants in the plant factory. A profitability analysis of applying air anions revealed that the annual net profit per 1500 m$^2$ cultivation area was about USD 60,000 and USD 70,000 for red leaf lettuce and Lollo bionda lettuce, respectively. Therefore, the application of air anions to lettuce in plant factories or greenhouses could increase crop production and has high economic feasibility.

**Keywords:** electric field; electrical stimulation; photosynthesis; crop productivity; commercial viability; growth improvement



## 1. Introduction

Electric potentials near the surface of Earth are approximately 150–300 V/m on average fair-weather conditions, and plants growing on Earth are exposed to these natural electric fields [1]. The ionosphere is the ionized part of Earth's upper atmosphere, from about 50 km to 1000 km, where air molecules are ionized by the solar energy. The layer of 70–140 km in the ionosphere has high conductivity and a small potential, so it acts as a perfect magnetic shield in the atmosphere. In the region lower than the shield, the atmosphere acts as a dielectric, creating a positive potential between Earth and the ionosphere, also called the electrosphere. The number and types of charged particles in the atmosphere depend on weather, season, climate, region, and other factors [2]. These differences affect the electrical condition of the atmosphere, resulting in a positive potential near the surface of Earth on sunny days, which reverses and becomes stronger on rainy days. In addition, the greater the environmental pollution, the larger the potential, whereas in an area with less environmental pollution, such as the sea, the potential is relatively low.

While the physiological effects of environmental factors such as light, temperature, and humidity on plant growth have been studied for a long time and the related literature is also quite extensive, the studies on the effect of electric fields on plants are still few [3]. However, as it has become increasingly clear that the electrical environment can affect plant growth, either positively or negatively, it is necessary to study the response of plants to electric fields in addition to other environmental factors.

Since plant cells repeatedly excrete and absorb ions through their membrane, charged ions exist in the cytosol, generating an electric field inside the plants [4,5]. In general, plants exposed to an electric field between Earth and the atmosphere have a small electric current. When exposed to an electric field of sufficiently high potential, such as a thunderstorm, a point discharge occurs, resulting in a relatively high current flow [6,7]. The effect of an electric current on plant growth is not yet completely understood, but experiments have been conducted in some natural and virtual environments. An artificially applied electric field can promote plant growth, as confirmed by studies performed since the early 20th century [1,3,8–14]. Since then, 'electro-culture', a technology that utilizes electricity for plant cultivation, has been employed mainly in greenhouses where environmental control is possible [15,16]. According to a previous study, a strong electrostatic field induces rotational movements and the translation of electric charges and dipoles and can also change the rate of chemical reactions, molecular binding forces, and the shape and structure of proteins [17]. Several studies on the effects of electric fields on plants have recently reported the promotion of plant growth [18–20]. Growth enhancement as well as the promotion of physiological processes such as photosynthesis and respiration were major phenomena resulting from the application of air anions in several plants [21–23]. However, further experiments are still needed to verify the effects of air anions, and studies on plant responses are currently lacking.

The global megatrends of increasing population, diminishing water supply, increased urbanization, and climate change have contributed to globally decreasing the stocks of arable land per person. Under these circumstances, the sustainability of the traditional farming model based on large rural farms is likely to come under threat in the coming decades [24]. One approach to solve this challenging problem is a plant factory, a closed growing system that enables to achieve a constant production of crops all year round. However, these plant factories have high starting costs, the production volumes are also not as large as those of broad-acre farming, and scaling-up may add costs and complexity. Therefore, it is inevitable to increase the profitability of plant factories through cultivation strategies that can improve crop production.

Leafy vegetables such as lettuce (*Lactuca sativa*) have been commonly grown in plant factories due to their short cultivation period and low input energy [25]. The application of air anions to lettuce in a controlled environment could contribute to improving the profitability of plant factories by not only increasing plant growth, but also shortening the plant growing period [15,16]. In this paper, this electric cultivation method was applied to an extensively used vegetable, lettuce; the growth-promoting effect was confirmed, and further analysis was made on whether this technique could be used commercially. It is anticipated that the information presented in this study will provide practical guidance towards improving the crop productivity of other similar leafy vegetables in plant factories.

## 2. Materials and Methods

### 2.1. Plant Materials and Culture Conditions

In order to establish an approach using air anions, an experiment was conducted to verify the effectiveness of air anions in a 15 m$^2$ cultivation area in an actual commercially operating plant factory, i.e., InsungTec Co., Ltd., Yongin, Republic of Korea. Two cultivars of lettuce (L. *sativa* cv. Lollo bionda *Multigreen* and red leaf cv. *Jeokchima*) were used as plant materials in this study. Lollo bionda lettuce is commonly served in leafy salads worldwide because of its light green, curly leaves that are very decorative and tender in taste. Red leaf lettuce has a high nutritional density and is popular as a leafy vegetable wrapped in meat in Korea. The seedlings grown for 14 days were transplanted into the nutrient film technique (NFT) system and cultivated under a 200 μmol·m$^{-2}$·s$^{-1}$ photosynthetic photon flux density (PPFD) provided for 16 h·d$^{-1}$ by fluorescent lamps (FDF32SS-EX-D, Osram, Seoul, Republic of Korea) inside the plant factory. The light intensity was measured at the top of the plant at 20 cm intervals in the cultivation area using a PAR sensor (LI-1400, LI-COR, Lincoln, NE, USA). The average air temperature and relative humidity in the plant

factory during cultivation were monitored and maintained by an environmental control system (InsungTec Co., Ltd.) at 22 °C and 70%, respectively. The EC and pH values of the nutrient solution (N/P/K = 17.3:4.0:8.0) were measured every 2 days using a pH·EC meter (WTW pH 3210, WTW GmbH, Weilheim, Germany) and adjusted to 1.3 dS·m$^{-1}$ and 6.0, respectively. The control and treatment groups for the two cultivars consisted of 3 plots each. Five hundred thirteen plants were grown in 5 rows of about 100 plants in a plot (15 m$^2$) at a planting distance of 15 cm. The positions of the plants were systematically rotated every day to minimize growth differences caused by imbalances of light and air anion distribution.

### 2.2. Application of Air Anions

According to our previous studies, we selected an air anion concentration effective to promote plant growth [23,24]. The lettuce plants were supplied with air anions at approximately $5 \times 10^5$ ions·cm$^{-3}$ using high-voltage air anion generators (TFB-Y49, Trumpxp, Shanghai, China) for 4 weeks after transplanting. The concentration of air anions was achieved by varying the number of the generators and measured at the plant level with an air anion counter (COM-3600, Com System, Tokyo, Japan). Thus, 32 generators were installed per 5 m$^2$, and 96 generators were used in the experimental area of 15 m$^2$. The plants receiving the treatment, as well as the control plants, were placed in separate sections of the chamber, and the generators were operated continuously throughout the day during the treatment period (Figure 1A).

### 2.3. Plant Growth

Fifteen lettuce plants per treatment were collected every two weeks of treatment. The plants were separated into shoots and roots and weighed using an electronic scale (Si-234, Denver Instrument, Bohemia, NY, USA). Both plant parts were oven-dried at 70 °C for 72 h to determine the dry weight. The leaf area was determined using a leaf area meter (LI-3100, LI-COR), and the leaf width and length were measured using a digital Vernier caliper (NA530-300S, Bluetec, Changwon, Republic of Korea). The leaf shape index (LSI) was calculated using the following equation: LSI = leaf length/leaf width (using the fifth leaf from the base) [26]. The specific leaf area (leaf area per unit dry mass; SLA) was calculated by dividing the measured leaf area by the dry weight. The chlorophyll content of the leaves was measured using a portable chlorophyll meter (SPAD-502; Minolta, Osaka, Japan).

### 2.4. Economic Efficiency Evaluation

Based on the results of the lettuce plants, the economic efficiency of applying air anions in plant factories was evaluated by a cost–benefit analysis. The net profit for one cropping period in 15 m$^2$ of cultivation area was calculated by subtracting the total expenses of applying air anions from the total revenue. The total revenue was calculated by multiplying the total yield by the retail price of lettuce. The price of lettuce used for the evaluation was based on the wholesale price provided by the Korea Agricultural Marketing Information Service (www.kamis.or.kr, accessed on 5 July 2022) at the time of the analysis. The total expenses included the cost of the air anion generators and the electric power consumed to generate the air anions during one cropping period. The energy consumption of the air anion generators was measured using a multimeter (FLUKE-115, Fluke, Everett, WA, USA) during the experiment. The total electric power consumed to control the growth conditions was equal between the control and treatment groups, and the only difference was the use of the air anion generators. The economic feasibility on a one-year basis was calculated by expanding the cultivation area to 1500 m$^2$ and multiplying the net profit per cropping season by the number of harvests available for one year.

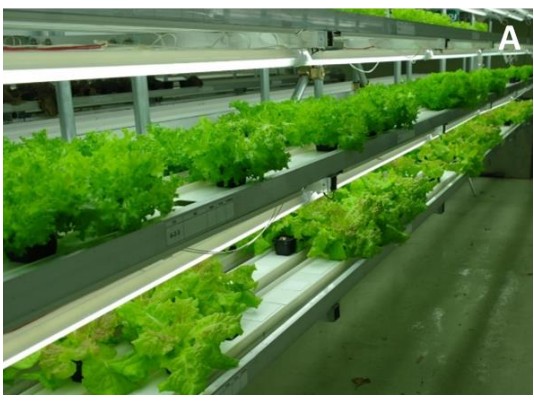

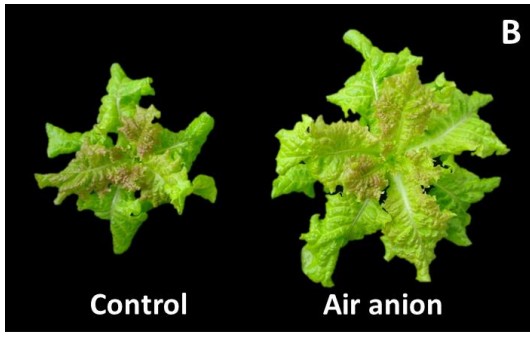

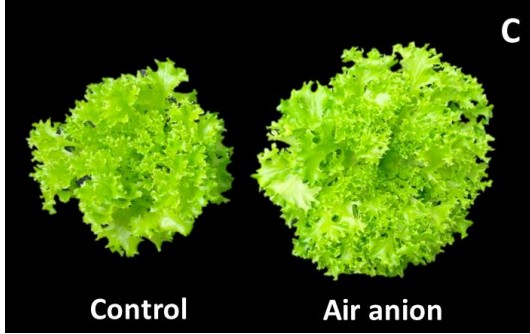

**Figure 1.** Application of air anions to lettuce cultivation in a commercial plant factory (**A**). Red leaf (**B**) and Lollo bionda lettuce (**C**) plants grown under the application of air anions for 28 days.

### 2.5. Statistical Analysis

The experiments were duplicated using different plants, and the plots were randomly assigned to each treatment. The growth parameters were verified in 15 plants by selecting 5 plants per plot per treatment. The data were analyzed using the Welch's t-test by the Statistical Package for the Social Sciences program (SPSS version 12.0, SPSS Inc., Chicago, IL, USA).

## 3. Results

### 3.1. Growth of Lettuce

The air anions induced remarkable differences in the growth of the lettuce plants (Figure 1B,C). The leaf number of red leaf lettuce was not increased by the air anions on the 28th day of treatment, the day of the final harvest, whereas the leaf area was significantly increased by approximately 1.3 times compared to the control (Figure 2A,B). The fresh weight of the shoots was also about 1.5 times greater than that of the control, and the dry weight was significantly increased by approximately 1.2 times (Figure 3A,B). Similarly, for the Lollo bionda lettuce, only the leaf area was increased by 1.5 times compared to the control (Figure 2C,D). The fresh and dry weights of the shoots were significantly increased in the anion-treated Lollo bionda lettuce on the 28th day by approximately 1.5 and 1.1 times,

respectively (Figure 3C,D). The SLA showed different results depending on the cultivar of lettuce: it was significantly decreased by 10% by air anion treatment in the red leaf lettuce and increased by 20% in the Lollo bionda lettuce (Figure 4).

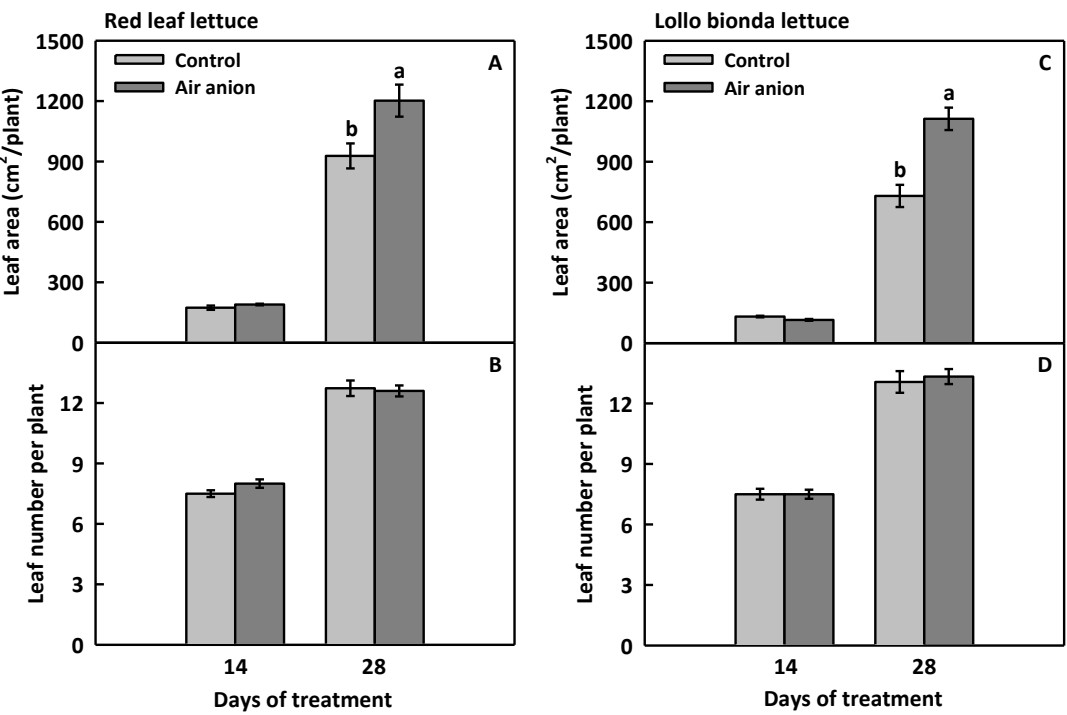

**Figure 2.** Leaf area (**A**,**C**) and number (**B**,**D**) of two types of lettuce grown under the application of air anions ($5 \times 10^5$ ions·cm$^{-3}$) for 14 and 28 days. Different lowercase letters indicate significant differences at $p < 0.001$ (n = 15).

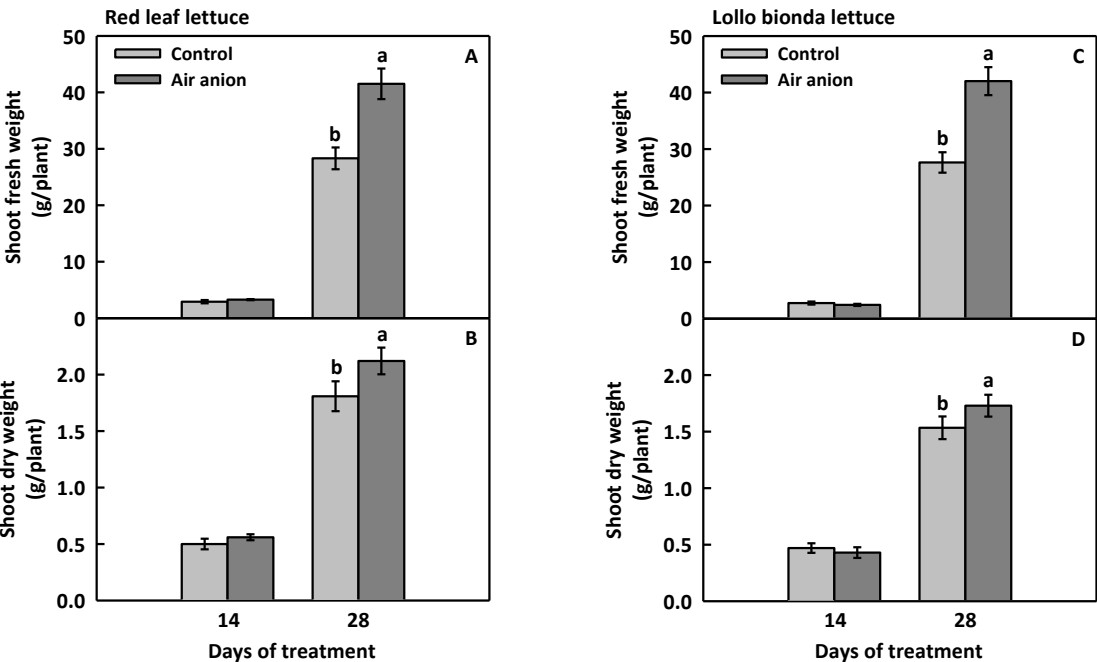

**Figure 3.** Shoot fresh (**A**,**C**) and dry weights (**B**,**D**) for two types of lettuce grown under the application of air anions ($5 \times 10^5$ ions·cm$^{-3}$) for 14 and 28 days. Different lowercase letters indicate significant differences at $p < 0.001$ (n = 15).

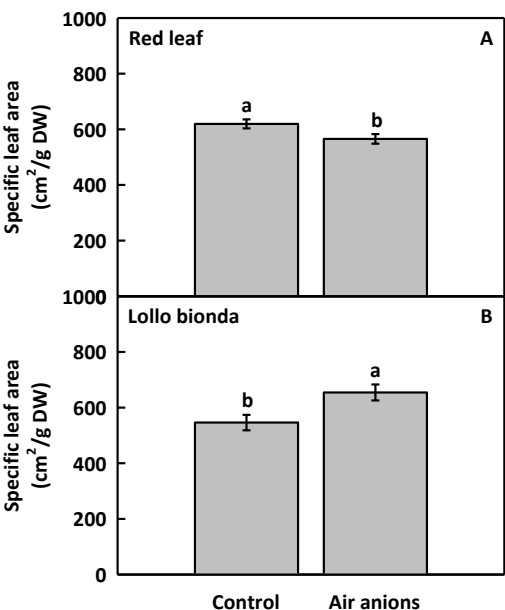

**Figure 4.** Specific leaf area of red leaf (**A**) and Lollo bionda lettuce (**B**) grown under the application of air anions (5 × 10⁵ ions·cm⁻³) at harvest. Different lowercase letters indicate significant differences at $p < 0.001$ (n = 15).

### 3.2. Chlorophyll Content in the Leaves

In order to estimate the change in photosynthetic ability of the two cultivars of lettuce plants under the application of air anions, the chlorophyll content in the leaves was measured after 14 and 28 days of treatment. The SPAD value indicating the chlorophyll content was not significantly different between the control and the air anion-treated plants for both lettuce cultivars (Figure 5).

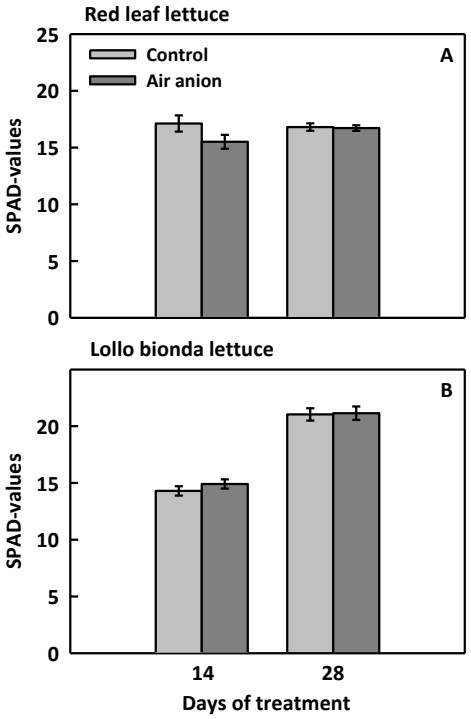

**Figure 5.** Chlorophyll content in the leaves of red leaf (**A**) and Lollo bionda lettuce (**B**) after the application of air anions for 14 and 28 days (n = 15).

### 3.3. Economic Efficiency of Applying Air Anions

Table 1 shows the results of the analysis of the expenses and revenues of applying air anions ($5 \times 10^5$ ions·cm$^{-3}$) to two cultivars of lettuce in the operating plant factory examined. Each of the 513 red leaf and Lollo bionda lettuce plants were grown for 4 weeks in one cropping season in a unit plot, and the profit and loss for one year were calculated. The cost of the air anion generators (96 units) used in the 15 m$^2$ area was USD 25, and the electric cost for the operation of the generators was about USD 1.3 (80.74 kWh; 1.63 US cent/1 kWh). As a result, the total expense for the air anion treatment in this experiment was computed to be USD 26.3.

**Table 1.** Expenses, yield, and revenue after applying air anions to the cultivation of two cultivars of lettuce in a plant factory.

| Type of Lettuce | Treatment | |
|---|---|---|
| | Control | Air Anion |
| Red leaf | | |
| Expenses (USD) | - | 26.3 |
| Yield (kg) | 14.5 | 21.3 |
| Revenue (USD) | - | 108.8 |
| Lollo bionda | | |
| Expenses (USD) | - | 26.3 |
| Yield (kg) | 14.2 | 21.6 |
| Revenue (USD) | - | 125.8 |

Cultivation period: 4 weeks (1 harvest); cultivation area: 15 m$^2$; a set of air anion generators installed per unit area (15 m$^2$); as expenses we considered those associated with the generators and consumed electricity; lettuce price: USD 1.6 and 1.7 per 100 g for red leaf and Lollo bionda lettuce, respectively; exchange rate applied as of 5 July 2022 (KRW 1300/USD 1).

The total production of red leaf lettuce grown in a 15 m$^2$ area for 4 weeks in the control and air anion treatment sections was 14.5 kg and 21.3 kg, respectively; thus, the production increased by about 6.8 kg due to the air anion treatment (Table 1). When the actual wholesale price of red leaf lettuce [USD 1.6 per 100 g] was applied to the increased production, the total revenue from red leaf lettuce production following the application of air anions was about USD 108.8/15 m$^2$. The expenses of applying air anions to Lollo bionda lettuce were the same as those calculated for the red leaf lettuce. The total production was 14.2 kg and 21.6 kg for the control and treatment groups, respectively, and the production was increased by about 7.4 kg as a result of the air anion treatment. Therefore, based on the wholesale price of green lettuce, the total revenue from Lollo bionda lettuce production was about USD 125.8/15 m$^2$.

After calculating the net profit by subtracting the expenses associated with the air anion treatment from the total revenue, the net profit resulting from the application of air anions for 4 weeks in a 15 m$^2$ area for red leaf lettuce was about USD 82.5, and for Lollo bionda lettuce was about USD 99.5. Based on these results, if the cultivation scale is expanded to 1500 m$^2$ (15 m$^2$ × 4 tiers × 25 lines) and the plants are cultivated seven times a year, the net profit would be USD 57,750 for red leaf lettuce and USD 69,650 for Lollo bionda lettuce (Table 2).

**Table 2.** Economic efficiency of applying air anions to the cultivation of two cultivars of lettuce in a plant factory.

| Type of Lettuce | Net Profit for Unit Area (15 m$^2$) | Net Profit for 1500 m$^2$ |
|---|---|---|
| Red leaf | USD 577.5 | USD 57,750 |
| Lollo bionda | USD 696.5 | USD 69,650 |

Cultivation period: 1 year (7 harvest); cultivation area: 15 m$^2$ × 4 tiers × 25 lines = 1500 m$^2$; lettuce price: USD 1.6 and 1.7 per 100 g for red leaf and Lollo bionda lettuce, respectively; exchange rate applied as of 5 July 2022 (KRW 1300/USD 1).

## 4. Discussion

Exposure to air anions accelerated the growth of lettuce, and a primary change was observed in the shoots rather than in the roots (Figures 1–3). Many studies have been conducted on the effects induced by electricity in plants, and a growth-promoting effect has been reported as the most representative response [18]. Electroculture is the practice of applying a strong electric field or a source of small air ions for growing plants [27]. In the 20th century, researchers investigated the effects of positive and negative ions in the air on barley (*Hordeum vulgare*), oat (*Avena sativa*), and garden cress (*Lepidium sativum*). It was found that the applied ions promoted crop growth [22,28,29]. In our previous studies, a growth-boosting effect was observed when air anions were applied to lettuce, kale (*Brassica oleracea* var. *acephala*), and spinach (*Spinacia oleracea*) [30–32], and the application of electric current directly to the rhizosphere in kale exhibited a similar effect [20]. These studies support our results showing the possibility of promoting the growth of lettuce plants by using air anions.

The air ion treatment has shown potential to cause positive changes in root growth. Previously, Smith and Fuller (1961) reported that an electric field formed by air cations promoted the biosynthesis of indole acetic acid (IAA) in *Microcoleus vaginatus* [33]. Furthermore, a study on tobacco (*Nicotiana tabacum*) callus suggested the possibility that a 1–2 $\mu$A direct current (DC) promoted the polar transport of IAA [34]. Our previous studies also confirmed that root development was markedly enhanced by air anions or electric fields in kale and spinach [20,30,32]. The findings of the current study suggest that the application of air anions may have increased the level of auxin in the plants, which moved easily to the roots and promoted root formation [35]. However, the relationship between the external electric fields and the biosynthesis and/or transport of auxin in plants should be verified through a molecular approach in further studies.

Our previous studies demonstrated that air anions or electric fields promote photosynthesis in lettuce, kale, and spinach [20,30–32]. Elkiey et al. (1985) found that exposure to air ions (cations $1 \times 10^5$ ions·cm$^{-3}$; anions $4 \times 10^5$ ions·cm$^{-3}$) improved photosynthesis and respiration [36]. Respiration was also promoted in *Arum maculatum*, common wheat (*Triticum vulgare*), and broad bean (*Vicia faba*) when the plants were exposed to an electrostatic field of 5–10 kV/m [37]. In other words, the electric fields generated by air anions promote representative metabolic reactions in plants, such as photosynthesis and respiration, thereby promoting the growth of shoots and roots. Negatively charged ions in the atmosphere form an electric field with surrounding positively charged ions. In the electric field artificially applied by air anions, cations in the leaf cells are likely to be attracted toward the negative electric field formed by the air anions [30]. The elevated number of cations, including potassium ions (K$^+$), could expand the guard cells in the leaf epidermal tissue, which in turn will open the stomata [38,39]. In our previous studies, the exposure to air anions enhanced photosynthesis through increased stomatal conductivity and improved CO$_2$ gas exchange [30–32].

An economic analysis was conducted to evaluate whether the cultivation method utilizing electricity could be used commercially, and the results were positive (Tables 1 and 2). We evaluated the method's profitability by applying the results of this experiment to a plant factory of 1500 m$^2$, currently in operation. The results revealed that the net profit resulting from the application of air anions for one year was approximately USD 60,000 for red leaf lettuce and USD 70,000 for Lollo bionda lettuce. The cost of an air anion generator was approximately USD 0.26; therefore, the installation cost and the electric power consumption were very low. In addition, the net profit presented in this study can remarkably increase as the cultivation scale increases. The method was applied to only a part of the space of the plant factory; if applied to the actual cultivation area of the plant factory, a much higher net profit could be obtained.

The increasing trend of yield differed depending on the plant species. When the plants were cultivated for one cropping season (4 weeks) using 96 air anion generators in a 15 m$^2$ area of the plant factory, the net increased production was 6.8 kg for red leaf lettuce and

7.4 kg for Lollo bionda lettuce (Table 1). Accordingly, the economic effect of the air anion treatment was found to be greater for red leaf than for Lollo bionda lettuce. Therefore, it is suggested that a plant species associated with high growth upon the application of air anions, a high market price, and a short cultivation period should be selected. The results of this study demonstrate that there is a close relationship between air anions and plant growth in both cultivars of lettuce, but further exploration is still required in other plant species. Thus, the approach and findings presented here show how electric field control through the supply of air anions can serve as a strategy to improve crop productivity and increase the economic efficiency of plant factories.

## 5. Conclusions

This study demonstrated that the application of air anions to lettuce cultivation could have a positive effect on plant growth and lead to a noticeable improvement in the profitability of plant factories. There has been a debate about the effect of electricity on plants for a long time. However, various research studies over the last decade have proved that electric fields are practically applicable as a new cultivation technology to increase crop production. This current study suggests that electric fields should be considered as an important environmental factor affecting plant growth. Applying air anions or electric fields to crop cultivation could be a simple and economical method to improve plant growth. In addition, the time required for the crop to reach the saleable stage can be shortened, resulting in higher profits. Further studies should elucidate the molecular and cellular mechanisms by which electricity affects plant growth, leading to a consistent plant response to electricity.

**Author Contributions:** S.L.: conceptualization, investigation, formal analysis, visualization, writing—original draft preparation. M.-J.S.: investigation. M.-M.O.: conceptualization, writing—review and editing, project administration, funding acquisition. All authors have read and agreed to the published version of the manuscript.

**Funding:** This research was supported by the Basic Science Research Program through the National Research Foundation of Korea (NRF) funded by the Ministry of Education (grant number 2020R1I1A3074865).

**Conflicts of Interest:** The authors declare that they have no known competing financial interest or personal relationships that could have appeared to influence the work reported in this paper.

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
