# Peer review of "Effects of Air Anions on Growth and Economic Feasibility of Lettuce: A Plant Factory Experiment Approach"

_sustainability, doi:10.3390/su142215468_

Round 1

Reviewer 1 Report

·        Abstract lacks methodology, the methodology should be comprehensively described in the abstract

Reviewer 2 Report

Thank you your effort. Here are specific points need clarifications as follows;

Line 1-2: 'Growth Acceleration Effects of Air Anions in Endive and Lettuce, and Economic Feasibility of Their Application' should be 'Effects of air anions on growth and economic feasibility of endive and lettuce: A plant factory experiment approach'.

Line 28-74: Please add more background information on endive and lettuce production systems in plant factory environment as well as key research questions with respect to 'sustainability of the production systems'.

Line 80-102: Reasons and purposes for using endive and lettuce in the experiment? Are you trying to both crops under various anion treatments? Any relations to 'sustainability' of both crop production systems in Korea?

Line 85: How many plants in a chamber? How many plants used for the final yield assessment?

Line 104-113: Reasons for using three levels of air anions and only one anion level on Lettuce?

Line 121: please provide specific aerial plant parts of endive and lettuce.

Line 132-138: Why are you providing only details of photosynthetic rate measurements on endive? What about lettuce photosynthetic rate measurements?

Line 140-154: Why are you providing only details on lettuce?

Line 176B: What is the different of the top and the bottom pairs (Control and Air Anion) on lettuce.

Line 178: Figure 3B, the Y-axis unit 'cm square' per what? Figure 3C and 3D, the Y-axis unit is gram per what?

Line 180: Figure 4 the Y-axis unit is gram per what?

Line 192: Typo, 'devreased' should read 'decreased'.

Line 195: Figure 5A and 5C, the Y-axis unit 'cm square' per what? Figure 5B and 5D, 'Leaf number' per what?

Line 199: Figure 6A and 6C, the Y-axis unit 'g' per what? Figure 5B and 5D, shoot dry weight 'g' per what?

Line 220-249: In section 3.3, heading should read 'Lettuce economic efficiency...' as no results/analysis on endive was provided.

Line 250-303: Discussion should include 'limitations' of your research.

Line 302-303: 'leafy vegetables' is claiming a larger domain than the two, endive and lettuce, crops that were used in the study.

Line 305: Please include 'a debate' about the effect of electricity on plants in your literature review section and elaborate in the discussion section. Any issue related to sustainability of crop production, please be specific?

Line 305-314: Please conclude based on your experimental results from endive and lettuce under plant factory environments. 

Line 315: should read 'elucidate the molecular and cellular mechanisms by which electricity affects plant growth,'.

Reviewer 3 Report

Define at the end of the introduction the objective of the research

Authors should include a hypothesis

Separate °C from the immediately preceding number

Include the model of the growth chamber

Include in the introduction the physiological principles of the method used on plants

Why was the control carried out in one place and the treatments in another?

In materials and methods: in paragraph one the authors describe the use of a growth chamber, then in paragraph two they describe another site, but the relationship between one site and the other is not understood, specify that point.

Put the same nomenclature inside figure 2 a and b                        

In the foot of figure 1 should specify the treatments

Why were some variables measured on two different species and other variables on two different varieties of the same species (lettuce)? This was not specified in the methodology

Line 205: anions per anons

I don't understand why some variables are measured in one type of lettuce and other variables in another type (Figure 8)

Legend of Figure 8, the units should be µmol m-2 s-1 and SPAD-values or SPAD-units

Lines 270 and 271. You can't argue that, because you didn't test the auxins

Figure 3 and 4 only endive; Figure 5, 6 and 7 only lettuce; Figure 8 photosynthesis rate only in endive, but SPAD-values in all genotypes. Table 1 and 2 economic analysis only in lettuce. Why were not all variables measured in all genotypes?

The research is interesting, but there is no order to compare or see the effects. I suggest restructuring the order of the results, even they could remove endive and stay only with the lettuce results, but that consistency be seen between the methodology, the results and the discussion, based on a hypothesis and a well-specified objective from the last paragraph of the introduction . If you decide to include endive and lettuce, I suggest you present the same variables in both species.

Round 2

Reviewer 2 Report

Thank you for efforts to revise the manuscript. Here are some specific comments.

Line 20: 'leafy vegetables' should read 'lettuce', based on your data set.

Line 63: 'metabolism' are not included in the results. 

Line 78-79: Need to provide a reference for 'increasing their growth' and 'shortening their growing period.

Line 83: 'of leafy vegetables' should read 'of other similar leafy vegetables'.

Line 86-96: Experiment with replications?

Line 88-89: Reasons for selecting this two cultivars, with respect to growth and/or economic value?

Line 108-109: Figure 1 B: the labels and the captions are wrong.

Line 149: Fig. not Figs.

Line 175: Why n=10 not n=15 as in Figures 2-4?

Line 218: Please remove 'endive' and please be careful.

Line 179: 513 plants based on 15 square meter of your Plant Factory experiment?

Line 221: No 'endive' in Fig 4. Please be careful.

Line 280: You have no experimental results (Figures 2-5) to support this line of conclusion.

Reviewer 3 Report

Congratulations to the authors, they did a good job with the corrections. 

Author Response

We appreciate the time and effort that you have dedicated to providing your valuable feedback on our manuscript. Thank you very much.

Round 3

Reviewer 2 Report

To Authors:

Thank you for efforts to improve the manuscript to meet the Journal's quality. However, I find that some issues remain to be clarified as follows:

1. Line 143: Please provide details of the validation setting and results.

2. Line 163-164: Figure 2 caption must be on the same page with the Figure.

------
